# Neuropharmacological Activities of *Ceiba aesculifolia* (Kunth) Britten & Baker f (Malvaceae)

**DOI:** 10.3390/ph15121580

**Published:** 2022-12-18

**Authors:** Chrystyan Iván Bustos-Gómez, Deisy Gasca-Martínez, Eunice Yáñez-Barrientos, Sergio Hidalgo-Figueroa, Maria L. Gonzalez-Rivera, Juan Carlos Barragan-Galvez, Juan Ramón Zapata-Morales, Mario Isiordia-Espinoza, Alma Rosa Corrales-Escobosa, Angel Josabad Alonso-Castro

**Affiliations:** 1Maestría en Ciencias Farmacéuticas, División de Ciencias Naturales y Exactas, Universidad de Guanajuato, Guanajuato 36200, Mexico; 2Unidad de Análisis Conductual, Instituto de Neurobiología, Campus UNAM-Juriquilla, Juriquilla 76230, Mexico; 3Departamento de Química, División de Ciencias Naturales y Exactas, Universidad de Guanajuato, Guanajuato 36200, Mexico; 4CONACyT-División de Biología Molecular/Instituto Potosino de Investigación Científica y Tecnológica A. C., San Luis Potosí 78216, Mexico; 5Departamento de Farmacia, División de Ciencias Naturales y Exactas, Universidad de Guanajuato, Guanajuato 36200, Mexico; 6Instituto de Investigación en Ciencias Médicas, Departamento de Clínicas, División de Ciencias Biomédicas, Centro Universitario de los Altos, Universidad de Guadalajara, Tepatitlán de Morelos, Jalisco 47620, Mexico

**Keywords:** anxiolytic-like activity, antidepressant-like activity, traditional medicine, adrenergic system, myristic acid

## Abstract

*Ceiba aesculifolia* (Kunth) Britten & Baker f (Malvaceae) is used for the folk treatment of mood disorders. *C. aesculifolia* bark was extracted in ethanol, and the extract (CAE) was chemically standardized using gas chromatography–mass spectrometry (GC-MS). This study evaluated the effects of CAE (10–100 mg/kg p.o.) on anxiolytic-like activity, sedation, locomotor activity, depression-like activity, and spatial working memory using in vivo rodent models. A possible mechanism for the anxiolytic-like and antidepressant-like actions induced by CAE was assessed using neurotransmission pathway inhibitors. Myristic acid was one of the compounds found in CAE using GC-MS. This study also evaluated the anxiolytic-like activity and the sedative actions of myristic acid and assessed a possible mechanism of action using neurotransmission pathway inhibitors and an in silico analysis. CAE elicited anxiolytic-like activity and antidepressant-like effects (ED_50_ = 57 mg/kg). CAE (10–100 mg/kg) did not affect locomotor coordination or induce sedation. The anxiolytic-like and antidepressant-like actions of CAE were reverted by prazosin, suggesting a possible participation of the noradrenergic system. The anxiolytic-like activity of myristic acid was reverted by the co-administration of prazosin and partially reverted by ketanserin. The docking study revealed that myristic acid can form favorable interactions within 5-HT2A and α1A-adrenoreceptor binding pockets.

## 1. Introduction

The study of medicinal herbs involves a multidisciplinary approach including the participation pharmacology, chemistry, medicine, analytical chemistry, molecular biology, ecology, sociology, etc. Traditional knowledge is a key element to isolate bioactive compounds for finding new drugs [1].

*Ceiba aesculifolia* (Kunth) Britten & Baker f (Malvaceae), commonly known as the “pochote” tree, is native to Mexico and is distributed from Mexico to Central America. *C. aesculifolia* grows in shallow soils in warm climates from sea level to 550 m of elevation and can be found in thorny forests and deciduous and sub-deciduous tropical forests. This tree can also grow in urban areas [2]. *C. aesculifolia* is 5 to 18 m tall, its trunk has thick thorns, and the bark is grayish brown. The fruits are large capsules containing many seeds, and the leaves are made of 5–7 leaflets. It has white, large, and showy flowers. Flowering time is from March to July and from September to January [3]. Bats are the main pollinators of this tree.

In Mayan culture, *C. aesculifolia* (Ya’ax ché) was considered a sacred tree responsible for holding the universe. According to Mayan mythology, the gods planted four enormous ceiba trees in each of the cosmic points of the universe, representing the Axis Mundi [3]. The bark and aerial parts of this tree are used for the empirical treatment of mental disorders, such as anxiety and depression [3,4], and its seeds and roots are used to treat gastritis, diabetes, kidney disorders, and skin infections [2,5]. Myristic acid, a fatty acid found in many plants, is used as a flavor adjuvant and food additive. Myristic acid has shown antidiabetic, antioxidant, antinociceptive, and anti-inflammatory activities [6,7]. The pharmacological properties of this tree species have not been scientifically documented. Therefore, this work aimed to evaluate the anxiolytic-like, sedative, locomotor, antidepressant-like, and spatial working memory actions of an ethanol extract of *C. aesculifolia* (CAE) bark using in vivo mouse models. In addition, the anxiolytic-like activity of myristic acid, a compound found in CAE, was evaluated with two in vivo models, and a possible mechanism of action was assessed using a docking study.

## 2. Results

### 2.1. Chemical Composition of C. aesculifolia Extract

Table 1 shows the list of possible compounds found in CAE by gas chromatography–mass spectrometry (GC–MS). This work identified 40 compounds (Table 1) classified as carbohydrates and derivatives (6, 7, 9, 10, 16–19, 23, 24, 28, 37, and 39), fatty acids and derivatives (1, 20, 22, 29, 31, 32, 33, 34, and 40), polyalcohol compounds (2, 3, 11, 12, 15, and 27), heterocyclic compounds (25, 35, 36, and 38), phenolic compounds (8, 14, and 26), a carboxylic acid (21), cyclic polyol (30), inorganic compounds (13), alkane (5), and γ-lactone (4). According to the gas chromatogram (Figure 1), the main components in CAE were 11, 16–19, 23, 24, 27–29, and 39.

### 2.2. Acute Toxicity

The acute toxicity of CAE resulted in LD_50_ higher than 2000 mg/kg per os (p.o.). Mice showed no signs of toxicity for 14 days. No signs of toxicity, such as immobility, piloerection, anorexia, unusual respiratory pattern, etc., were observed in mice.

### 2.3. Anxiolytic-Like Activity

CAE did not elicit anxiolytic-like activity in the light/dark test (Figure 2), but 1.5 mg/kg clonazepam (CNZ) increased (*p* < 0.05) the amount of time spent in the light compartment, the light/dark latency, and the number of crossings to the light compartment (Figure 2).

In the open field test, all doses of CAE increased (*p* < 0.05) the distance crossed in central squares (Figure 3a), whereas only 100 mg/kg CAE increased the time spent in the central squares (Figure 3b). Moreover, CAE (10–100 mg/kg p.o.) did not affect the total distance traveled (Figure 3c). The anxiolytic-like activity induced by CAE was not comparable to that induced by 1.5 mg/kg CNZ (Figure 3). The positive control 1.5 mg/kg CNZ showed anxiolytic-like activity but decreased the total distance traveled compared to the vehicle group (Figure 3c).

In the elevated plus maze test, CAE (10–100 mg/kg p.o.) increased (*p* < 0.05) the number of entries into open arms (Figure 4a) and decreased (*p* < 0.05) the number of entries in closed arms (Figure 4b), compared to the vehicle group. The anxiolytic-like effect of CAE (10–100 mg/kg p.o.) on the number of entries in open arms was comparable to that of 1.5 mg/kg CNZ (Figure 4a). Similarly, CAE (10–100 mg/kg p.o.) increased (*p* < 0.05) the time spent in open arms (Figure 4c) and decreased (*p* < 0.05) the time spent in closed arms (Figure 4d).

CAE induced anxiolytic-like action in the exploratory cylinder test. However, this effect was not in a dose-dependent manner (Figure 5a). All doses of CAE increased (*p* < 0.05) the number of head dips compared to the vehicle, but this effect was not dose-dependent (Figure 5b). CNZ (1.5 mg/kg) decreased (*p* < 0.05) the number of rearings (Figure 5a) and increased (*p* < 0.05) the occurrence of head dipping compared to the vehicle (Figure 5b). The anxiolytic-like activity of CAE was not comparable to that shown by the positive control in both tests (Figure 5a and 5b). Prazosin (0.05 mg/kg) attenuated (*p* < 0.05) the anxiolytic-like activity of CAE in the hole-board test (Figure 5c), whereas the pre-treatment with flumazenil or ketanserin did not affect the anxiolytic-like activity of CAE (Figure 5c).

### 2.4. Antidepressant-like Activity

CAE reduced (*p* < 0.05) immobility time in a dose-dependent manner in the tail suspension test (Figure 6a). In the same test, 100 mg/kg CAE showed a similar effect compared to 20 mg/kg fluoxetine (Figure 6a).

In the tail suspension test, only 100 mg/kg CAE decreased (*p* < 0.05) immobility time (Figure 6b). The antidepressant-like activity of CAE was higher in the tail suspension test when ED_50_ = 57 mg/kg than when ED_50_ > 200 mg/kg. The antidepressant-like actions of CAE were only blocked with the co-administration of 0.05 mg/kg prazosin (Figure 6c). Atropine or cyproheptadine did not block the antidepressant-like actions of CAE.

### 2.5. Effects on Sedation and Motor Coordination

None of the CAE doses induced sedation or affectations in locomotor activity. CAE (10–100 mg/kg p.o.) did not decrease the onset of sleep or extend the duration of sleep when compared with the vehicle group (Figure 7). CNZ (1.5 mg/kg) decreased (*p* < 0.05) the onset of sleep (Figure 7a) and increased (*p* < 0.05) sleep duration (Figure 7b) compared to the vehicle group. In addition, 1.5 mg/kg CNZ interfered (*p* < 0.05) with the locomotor activity of mice at 60 and 120 min post-treatment (Figure 7c). 

### 2.6. Effects on Spatial Working Memory

CAE (10–100 mg/kg) did not affect the number of arm entries or alternation behavior in the Y-maze test (Figure 8) compared to the vehicle group.

### 2.7. Myristic Acid Induces Anxiolytic-Like Effects

Myristic acid increased the number of head dips and dose-dependently decreased the number of rearings (ED_50_ = 3 mg/kg p.o.) compared to the vehicle group, which suggests anxiolytic-like actions (Figure 9a,b). This anxiolytic-like effect in both models was not comparable to 1.5 mg/kg CNZ. Myristic acid showed a dose-independent anxiolytic-like activity in the hole-board (Figure 9a). Pre-treatment with 0.05 mg/kg prazosin attenuated (*p* < 0.05) the anxiolytic-like activity of myristic acid in the hole-board test (Figure 9c) and the cylinder exploratory test (Figure 9d). Pre-treatment with 1 mg/kg ketanserin partially affected the anxiolytic-like activity of myristic acid, whereas the pre-treatment with 1 mg/kg flumazenil did not affect the anxiolytic-like activity of myristic acid in either model (Figure 9c,d).

### 2.8. Myristic Acid Induces Sedative Effects

Myristic acid did not alter the onset of sleep, whereas this compound prolonged (*p* < 0.05) the duration of sleep in a dose-independent manner (Figure 10b).

### 2.9. Docking Study with Myristic Acid

The possible molecular details responsible for the pharmacological effects of myristic acid were investigated using molecular docking studies against the 5-HT2A receptor and α1A-adrenergic receptor. The method was validated by superimposing both re-docked and co-crystallized ligand risperidone into the 5-HT2A receptor. The results suggest that myristic acid can form favorable interactions within 5-HT2A and α1A-adrenergic receptor binding pockets.

The overview of the docking complex of myristic acid and 5-HT2A is shown in Figure 11A,B. Ser 239 and Asp343 play important roles in the binding interaction through hydrogen bond formation. The carboxylic acid formed a polar interaction with mentioned residues and several hydrophobic interactions with Val156, Ile163, Trp336, Phe339, and Phe340. The free binding energy of myristic acid was −6.2 kcal/mol and −11.2 kcal/mol to re-docked risperidone. However, risperidone with the 5-HT2A antagonist maintained a strong interaction with Asp155, whereas myristic acid was anchored in Ser239 residue.

The molecular docking generated a complex between myristic acid and the α1A-adrenergic receptor model, which is shown in Figure 11C,D. The myristic acid interacts through van der Waals forces with Trp102, Ala103, Ala104, Val107, Phe308, Lys309, Phe312, and Tyr316 with a free binding energy of −6.2 kcal/mol.

## 3. Discussion

To our knowledge, this is the first report describing the chemical composition of an ethanol extract of *Ceiba aesculifolia* bark using GC-MS. A phytochemical screening using colorimetric methods revealed the presence of terpenoids, steroids, and triterpenoids in hexane and dichloromethane extracts of *Ceiba aesculifolia* bark [8]. This study revealed the abundant presence of carbohydrates and fatty acids and their derivatives in the ethanol extract of *Ceiba aesculifolia* bark. The difference in the chemical composition among reports relies on the polarity of the solvent. An HPLC analysis showed the presence of lepidine, capsaicin, and diphenylamine in a dichloromethane extract of Ceiba pentandra bark [9], whereas a GC-MS study showed that 3-ethyltetracosane, n-nonacosane, and n-docosane were the main components of a methanol extract of *Ceiba pentandra* aerial parts [10]. Linoleic acid, palmitic acid, and malvalic acid were the main components found by GC-MS in a Ceiba speciosa seeds oil [11].

Preclinical toxicity studies with medicinal plants are essential for obtaining information to avoid toxicity in humans [12]. The seeds of the related species *Ceiba pentandra* might cause dermatitis, whereas *Ceiba aesculifolia* is not considered a toxic tree [12]. The acute toxicity test revealed that the LD_50_ was higher than 2000 mg/kg, indicating low toxicity of CAE. Based on these findings and preliminary results obtained in our laboratory, the doses used for CAE ranged from 10 to 100 mg/kg p.o.

The light/dark test relies on the aversion of rodents to brightly lit open spaces, which mice avoid exploring [13]. In the open field test, rodents tend to walk near corners when exploring new environments. An increase in the time spent and distance covered in central squares represents anxiolytic-like behavior [14]. The elevated plus maze is based on the innate avoidance of rodents for unprotected, bright, and elevated places [15]. The cylinder exploratory test and the hole-board test can evaluate anxiogenic and anxiolytic behavior in rodents. A low number of rearings and an increase in head dips in the cylinder exploratory test and the hole-board test, respectively, indicate anxiolytic-like behavior, whereas an increase in rearings and an attenuation in head dips represent an anxiogenic behavior [16,17].

CAE showed anxiolytic-like activity in four of the five models. In the open field test and the elevated plus maze test, the anxiolytic-like activity of 50 and 100 mg/kg CAE was similar or comparable to that of 1.5 mg/kg CNZ.

The tail suspension and the forced swimming tests, used for screening possible antidepressant drugs, create an inescapable, stressful situation for mice that produces immobility [18,19]. The immobility of mice is an indication of behavioral despair. In the tail suspension test, no hypothermia is induced, and rodents recover motor activity immediately after the test. The tail suspension test presents more pharmacological sensitivity than the forced swimming test [15,16]. CAE showed higher antidepressant activity in the tail suspension test (ED_50_ = 57 mg/kg) than in the forced swimming test (ED_50_ > 200 mg/kg).

Pentobarbital induces sedative and hypnotic activity by binding to gamma-aminobutyric acid (GABA) A subtype receptors [20]. The potentiation of pentobarbital-induced sleep indicates hypnotic sedative activity [20]. CAE showed no sedative activity.

Antidepressant drugs such as fluoxetine and anxiolytic drugs such as CNZ reduce locomotor activity [18]. Unlike common antidepressant and anxiolytic drugs, CAE (10–100 mg/kg p.o.) did not induce sedation (evaluated with the pentobarbital-induced sleep test) or affect locomotor activity (assessed with the open field test and the rotarod test) in mice. Tests evaluating locomotor activity might assess hypo- or hyperactivity due to pharmacological treatment.

Dysregulation of norepinephrine levels in the locus coeruleus might lead to mood disorders such as depression and anxiety [21]. Prazosin, an α1 blocker, reverted the anxiolytic and antidepressant actions of CAE. Brain α1-adrenoceptors are involved in motor and exploratory activity; the blockade of these receptors abolishes movement and promotes anxiogenic activity in rodents [22]. The α1-adrenoceptors produce depolarization in neurons with the subsequent release of catecholamines, a target in the treatment of depression [22]. The results indicated that CAE did not affect locomotor activity, suggesting that CAE contains compounds that might act as α1 agonists.

The Y-maze alternation task evaluated the effect of acute CAE administration on spatial working memory. Spontaneous alternation is driven by rodents’ innate curiosity to explore novel environments. This model assesses spatial working memory depending on the interaction between different brain regions, such as the hippocampus and the prefrontal cortex [23]. CAE, in acute administration, did not influence spatial working memory in mice. We do not discard that chronic administration of *Ceiba aesculifolia* could increase spatial working memory in rodents.

Myristic acid was one of the compounds found in CAE. Previously, myristic acid administered in rats decreased the activity time and induced anxiolytic-like actions in the elevated plus maze without affecting motor coordination [24]. These previous findings were corroborated by the results found in this study, which showed that myristic acid induced anxiolytic-like activity in two murine models. In addition, the results also showed that myristic induced moderate sedative effects. The anxiolytic-like activity of myristic acid was abolished by the pre-treatment with prazosin, an α1-adrenergic receptor antagonist, and partially abolished by the pre-treatment with ketanserin, a 5-HT2A receptor antagonist. These findings suggest the possible participation of α1-adrenoceptors and 5HT2 receptors in the pharmacological actions of myristic acid. This possibility was assessed using an in silico study, which showed that myristic acid presents affinity for the 5-HT2A receptor and α1A-adrenergic receptor.

The 5-HT2A receptor is involved in the sleep–wake cycle and anxiety [25]. The reintroduction of the 5-HT2A receptor in the cortex of knock-out mice decreased their anxiogenic behavior [21]. Antagonists of the 5-HT2A receptor improve sleep maintenance and sleep efficiency without inducing motor incoordination [26]. α1-adrenergic receptors are important in the modulation of anxiety-like behavior [27]. The findings showed that myristic acid prolonged the duration of sleep in mice (present study) without affecting the locomotor coordination in rats, as previously reported by [24].

## 4. Materials and Methods

### 4.1. Drugs

Clonazepam, fluoxetine, polyvinylpyrrolidone (PVP), yohimbine, flumazenil, ketanserin, atropine, cyproheptadine, and prazosin were from Sigma-Aldrich (St Louis, MO, USA). The purity of these reagents was at least 95%, according to the manufacturer. Pentobarbital was from Pisa Farmaceutica (Mexico City, Mexico).

### 4.2. Plant Material

Samples of *Ceiba aesculifolia* were collected from the municipality of Irapuato (20.682935, −101.358535, state of Guanajuato, Mexico) in February 2021. Dr. Eleazar Carranza from the Isidro Palacios herbarium (SLPM) identified and preserved the plant material (voucher number 47019).

### 4.3. Extraction

*Ceiba aesculifolia* bark was dried at room temperature, ground, and extracted with ethanol for 10 days. The solvent was evaporated under reduced pressure.

### 4.4. GC-MS Analysis 

A Bruker 456-GC gas chromatograph coupled to a SCION TQ mass spectrometer (Bruker Daltonics) with a BP-5 MS column (30 m × 0.25 mm, 0.25 µm film thickness, SGE Agilent Technologies) was used. Operating conditions and the spectral analysis were carried out following the protocol described [28] with slight modifications. The flow rate of helium as a carrier gas was maintained at 1 mL/min, the injection volume was 1 µL, and the injection port was set to split 1:5 and maintained at 250 °C. The column oven temperature program was as follows: hold 100 °C initial temperature for 2 minutes, then ramp to 300 °C at 12°C/min and hold for 6 minutes (total run time was 24.6 minutes). Temperature settings for the transfer line heater and ion source of the mass spectrometer were 290 °C and 250 °C, respectively. Spectra were acquired in the *m*/*z* range 80–550; the filament was turned on after a solvent delay of 7 minutes; analysis was performed in “Full Scan” mode. A Bruker MS Workstation 8.0 was used for data collection, processing, and GC-MS control. 

The analysis of data was performed using the AMDIS software (Automated Mass Spectral Deconvolution and Identification System; http://www.amdis.net/) in “Use Retention Index Data” mode. The identification of compounds was made through MS Search 2.0 software using NIST and Wiley libraries.

### 4.5. Animals

Male CD-1 mice (weighing between 25 and 32 g) from the Institute of Neurobiology at the National Autonomous University of Mexico at Juriquilla, Queretaro, Mexico, were kept in polyacrylic cages with soft wood bedding under standard housing conditions with food and water ad libitum. The Institutional Research Ethics Committee for biomedical research projects of the University of Guanajuato (CIBIUG-P34-2019) approved all animal procedures.

### 4.6. Pharmacological Treatment

Trained observers performed the experiments in a silent room between 9 a.m. and 3 p.m. Different groups were assigned as follows (n = 8 mice per group): vehicle (saline solution and PVP 4:1), 1.5 mg/kg CNZ or 20 mg/kg fluoxetine (positive controls), CAE (10, 50, and 100 mg/kg p.o.), or myristic acid (0.1, 1, and 10 mg/kg p.o.). Each test was conducted 1 hour after treatment administration. All experiments were video recorded. Each apparatus was wiped with 70% ethanol at the end of each session.

### 4.7. Acute Toxicity

The procedure was conducted according to the protocol described [29]. Mice were treated with CAE (10–2000 mg/kg p.o.) and monitored for signs of toxicity and mortality every 24 hours for 14 days.

### 4.8. Light/Dark Test

One chamber was illuminated with white light and another chamber was dark. Each mouse, placed in the center of the light compartment, was allowed to move between the light and dark boxes via a door. The time and the of crossings to the light compartment and the light/dark latency were recorded for 5 min [13].

### 4.9. Open Field Test

The square bottom surface of the box was divided into 25 equal squares (Omnitech Electronics, Columbus, OH, USA). Each mouse was placed in the middle of the apparatus and allowed to explore. The distance crossed in the central squares, the time spent in the central squares, and the total distance traveled were recorded for 5 min [14].

### 4.10. Elevated Plus Maze

The apparatus consisted of two opposite open (35×6 cm) and closed (35×6 cm) arms joined by a central square (5×5 cm). The maze was raised about 30 cm above the ground. Each mouse was placed in the center of the apparatus, and the time spent in open arms and the number of entries into the open arms were recorded for 5 min [15].

### 4.11. Hole-Board Test

A flat closed platform (42 cm×42 cm×30 cm) with 16 spaced holes (3 cm in diameter) was used. The structure was 20 cm high. Mice were individually placed in the center of the platform, and the number of times each mouse dipped its head into the holes was counted for 5 min [17]. The anxiolytic-like mechanism of action of CAE was evaluated using 2 mg/kg flumazenil (a GABA antagonist), 1 mg/kg prazosin (an α1-adrenoreceptor blocker), or 1 mg/kg ketanserin (a 5-HT2 receptor blocker). Each inhibitor was intraperitoneally administered for 15 min before administering 10 mg/kg CAE. After 45 min, the hole-board test was carried out.

### 4.12. Exploratory Cylinder Test

Plexiglas cylinders were set in a vertical position. Each naïve mouse was put into the cylinder, and the number of rearings on its posterior limbs was counted for 5 minutes [16]. The anxiolytic-like mechanism of action of AOE was evaluated using 2 mg/kg flumazenil (a GABA antagonist), 0.05 mg/kg prazosin (an α1 adrenoreceptor blocker), or 1 mg/kg ketanserin (a 5-HT2A receptor blocker). Each inhibitor was intraperitoneally administered 15 minutes before administering 33 mg/kg AOE. After 45 minutes, the exploratory cylinder test was carried out.

### 4.13. Tail Suspension Test

Every mouse was suspended 40 cm above the ground by the tail fixed with adhesive tape [16]. Mice were considered immobile when they hung passively and completely motionless. The experiment lasted 6 min and the last 4 min were recorded.

### 4.14. Forced Swimming Test

Plexiglas cylinders were filled at 40% of their capacity with water at 25 °C. Immobility time was recorded for 6 min [18]. Mice were considered immobile when they remained floating passively with their nose above water. The experiment lasted 6 min and the last 4 min were recorded. Animals were pre-treated with prazosin at 0.05 mg/kg (an α1-adrenoceptor blocker), 1 mg/kg atropine (a muscarinic cholinergic antagonist), or 3 mg/kg cyproheptadine (H1 and 5-HT2B receptor antagonist) 15 min before the administration of CAE. After 45 min, the forced swimming test was carried out.

### 4.15. Pentobarbital-Induced Sleep Test

Mice were intraperitoneally injected with 50 mg/kg pentobarbital, a hypnotic dose, and the onset of sleep (from the injection of pentobarbital until the loss of locomotor activity) and the duration of sleep (loss of locomotor activity until the recovery of righting reflex) were registered [30].

### 4.16. Rotarod Test

Mice were trained daily for three days. Animals able to keep walking on the rotarod (Harvard Apparatus, Barcelona, Spain) set at 4 rpm for 4 min were selected for the experiment. The time spent on the rotarod was recorded at 60 and 120 min after treatment [31]. The cut-off time was 4 min.

### 4.17. Y-maze Alternation Task

This apparatus consisted of three equal arms (labeled as A, B, and C) at 120 degrees from each other [23]. Each mouse was placed at the beginning of one arm and allowed to explore the maze freely for 8 min. The sequence and number of arm entries were tracked for each mouse. An arm entry was counted when all four paws were placed inside the arm. Percent spontaneous alternation performance was calculated as [(number of alternations)/(total arm entries − 2)] × 100.

### 4.18. Homology Modeling

A model of the α1A-Adrenergic Receptor (α1A-AR) was generated with SWISS-MODEL (https://swissmodel.expasy.org/) using the α1B-Adrenergic Receptor (PDBid: 7b6w) which shared 62.81% sequence identity with α1A-AR. The amino acid sequence of the human α1A receptor was retrieved from uniprot.org (entry code P35348, ADA1A_HUMAN), and the model was subjected to refinement protocol of minimization process and 1 ns of molecular dynamic simulation (MD) conducted with a CHARMM 36 m force field, 0.15 mol/L salt concentration, and explicit water molecules. MD simulations were carried out by using NAMD [31]. This protocol was used to correct errors in structure (outliers and steric clashes).

### 4.19. Molecular Docking Studies

Molecular docking studies of myristic acid were carried out with 5-HT2A (PDBid: 6A93, resolution: 3.00 Å) and α1A-adrenoreceptor (homology model) structures, and conducted with the AutoDock Vina [32,33]. All of the water molecules and co-crystallographic ligand (risperidone) were removed from the 5-HT2A structure. Then, each hydrogen atom was added, non-polar hydrogen atoms were merged, and Gasteiger charges were assigned to all molecules (ligands and proteins, 5-HT2A and α1A-adrenoreceptor). Next, the torsions from compounds were allowed to rotate during the docking study. Each grid was centered at the crystallographic coordinates of risperidone (center_x = 14.231; center_y = −1.196 and center_z = 60.438) and (center_x = −1.837; center_y = −4.094 and center_z = −20.173) of 5-HT2A and α1A-AR, respectively. The grid dimensions were 18 × 14 × 14 points and 20 × 20 × 20 points with a default spacing of 5-HT2A and α1A-AR, respectively. Finally, the exhaustiveness employed was 20. All visualizations were obtained by using the program PyMOL v1.9 (The PyMol Molecular Graphics System, Version 1.9 Schrödinger, LCC.) and 2D interaction diagrams by Discovery Studio Visualizer, DSV v 21.1.

### 4.20. Statistical Analysis

Data were analyzed by one-way analysis of variance followed by Dunnett’s test. Results were considered statistically significant if *p* < 0.05.

## 5. Conclusions

CAE exhibited antidepressant-like (ED_50_ = 57 mg/kg) actions in the tail suspension test and anxiolytic-like activity with a non-dose-dependent effect in the hole-board, open field, and elevated plus maze tests. CAE did not induce sedative activity, nor did it alter the locomotor activity of mice. The anxiolytic-like and antidepressant-like effects of CAE are due to the possible participation of the adrenergic system. CAE, in acute administration, did not affect spatial working memory. Myristic acid showed anxiolytic-like activity (ED_50_ = 3 mg/kg) in the cylinder exploratory test. The docking study revealed that myristic acid can form favorable interactions within 5-HT2A and α1A-AR binding pockets.

## Figures and Tables

**Figure 1 pharmaceuticals-15-01580-f001:**
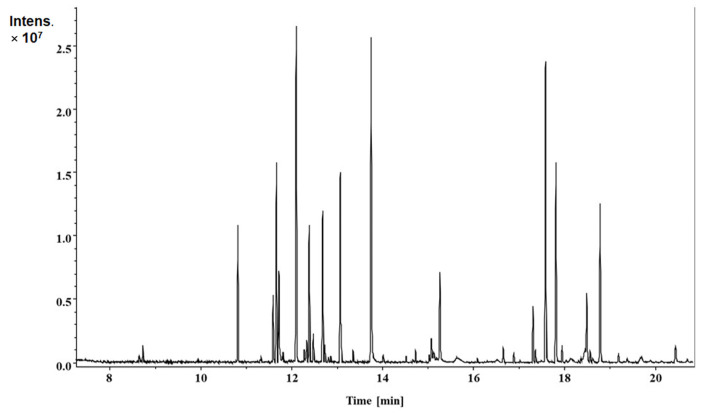
The chromatogram was obtained during GC-MS analysis of CAE. The tentative identification of the detected peaks corresponds to the data included in Table 1.

**Figure 2 pharmaceuticals-15-01580-f002:**
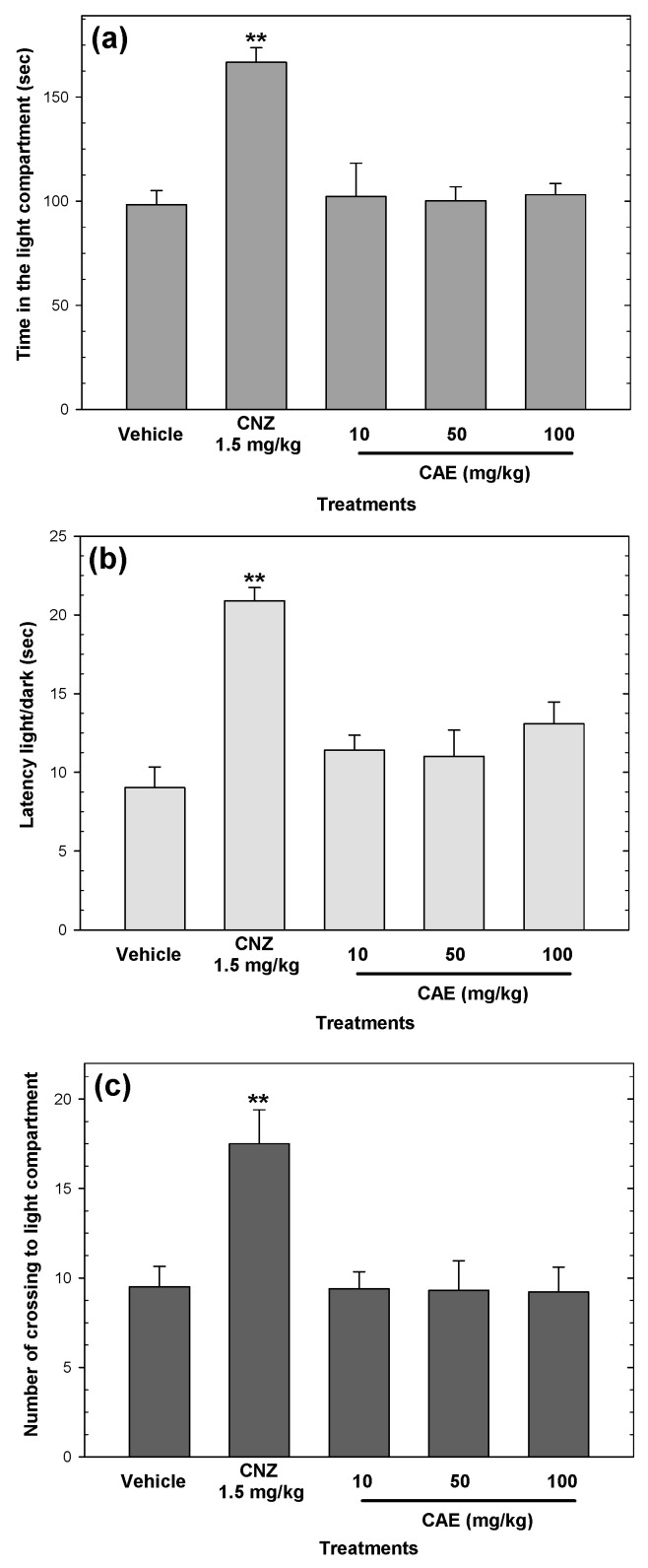
Anxiolytic-like actions of CAE (10–100 mg/kg p.o.) in the light/dark test were evaluated by the time spent in the light compartment (**a**), light/dark latency (**b**), and the number of crossings into the light compartment (**c**). Clonazepam (CNZ) at 1.5 mg/kg p.o. was used as the positive control. Bars represent mean values (± SEM) for the experimental group. n = 8, ** *p* < 0.05 compared to the vehicle group.

**Figure 3 pharmaceuticals-15-01580-f003:**
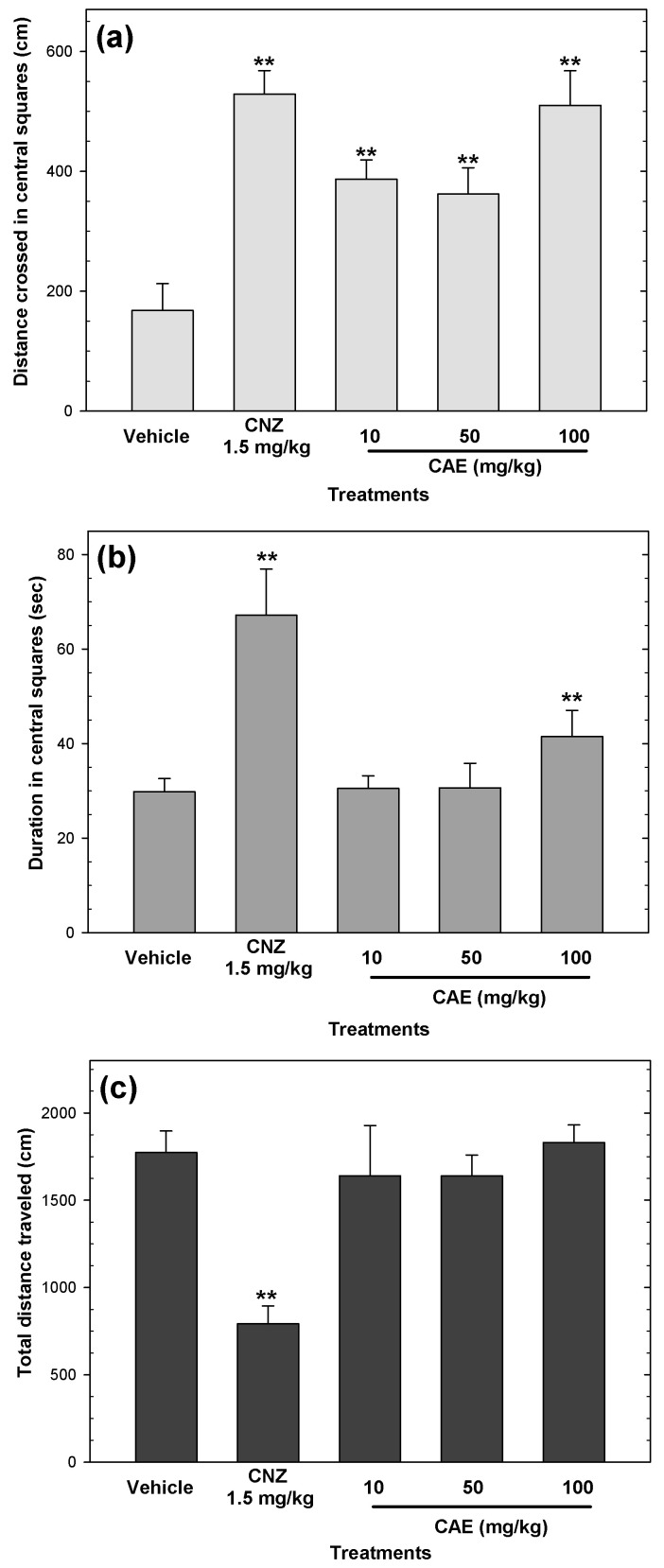
Anxiolytic-like actions and locomotor activity of CAE (10–100 mg/kg p.o.) in the open field test were evaluated by distance crossed in central squares (**a**), duration in central squares (**b**), and total distance traveled (**c**). Clonazepam (CNZ) at 1.5 mg/kg p.o. was used as the positive control. Bars represent mean values (± SEM) for the experimental group. n = 8, ** *p* < 0.05 compared to the vehicle group.

**Figure 4 pharmaceuticals-15-01580-f004:**
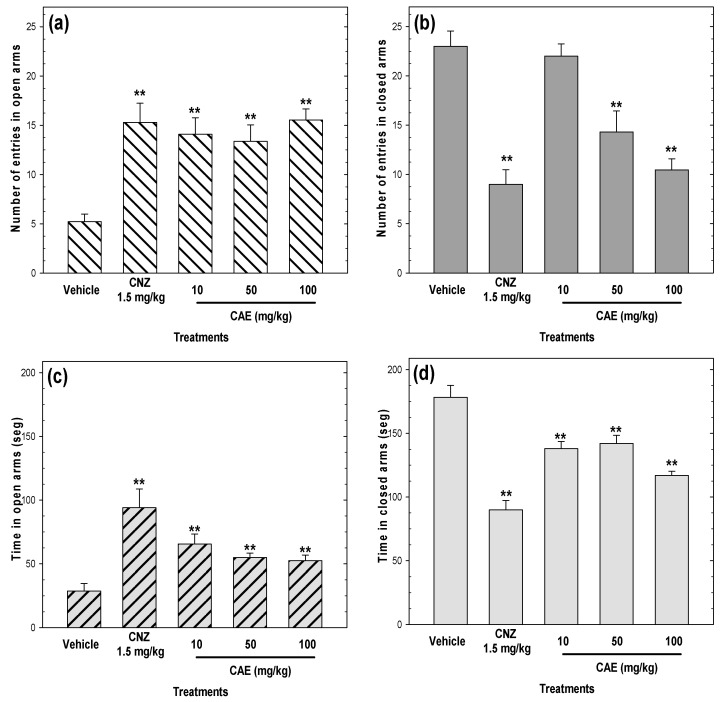
Anxiolytic-like effects of CAE (10–100 mg/kg p.o.) measured by the number of entries in open (**a**) and closed (**b**) arms and the time spent in open (**c**) and closed (**d**) arms and in the elevated plus maze during a 5 min exposure. Clonazepam (CNZ) at 1.5 mg/kg p.o. was used as the positive control. Bars represent mean values (± SEM) for the experimental group. n = 8, ** *p* < 0.05 compared to the vehicle group.

**Figure 5 pharmaceuticals-15-01580-f005:**
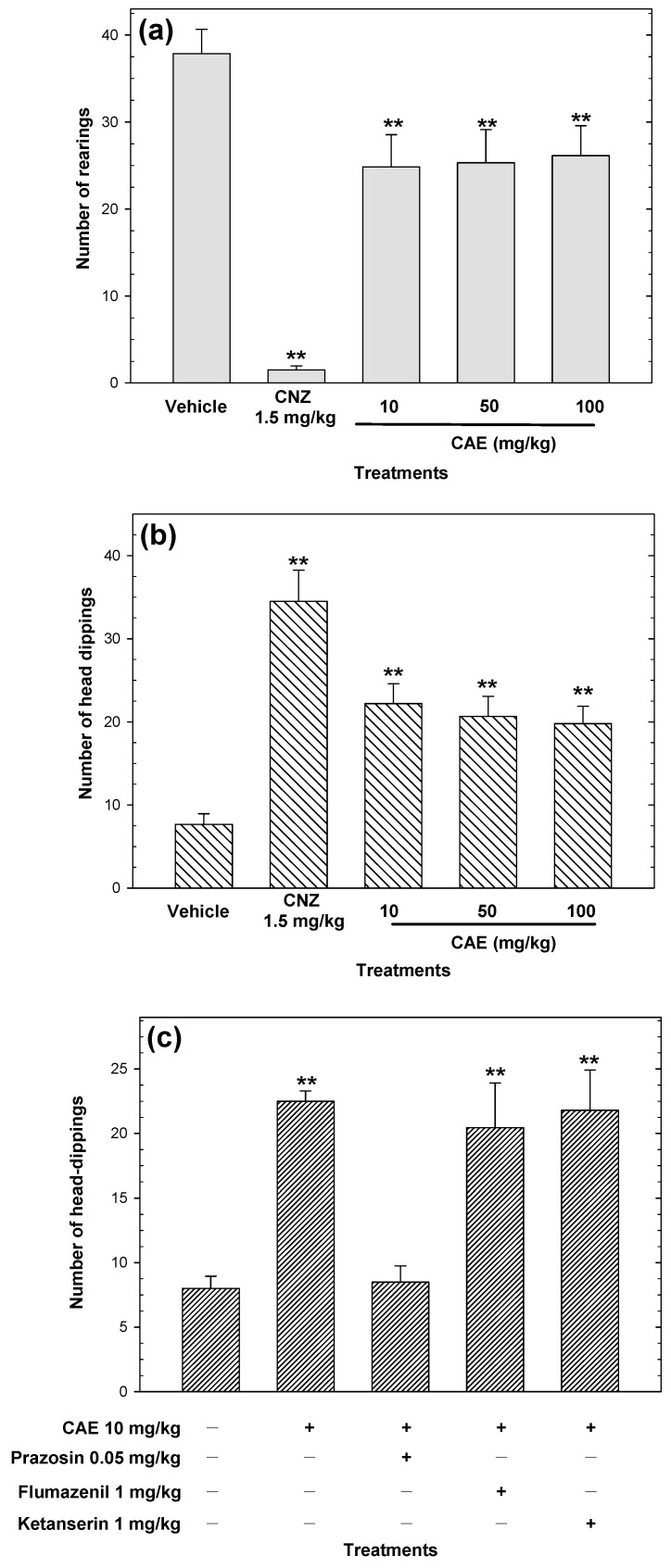
Anxiolytic-like effects of CAE (10–100 mg/kg p.o.) were measured in the cylinder exploratory test (**a**) and the hole-board test (**b**) during a 5 min exposure. The possible mechanism of action of CAE was evaluated in the hole-board test (**c**). Clonazepam (CNZ) at 1.5 mg/kg p.o. was used as the positive control. Bars represent mean values (± SEM) for the experimental group. n = 8, ** *p* < 0.05 compared to the vehicle group.

**Figure 6 pharmaceuticals-15-01580-f006:**
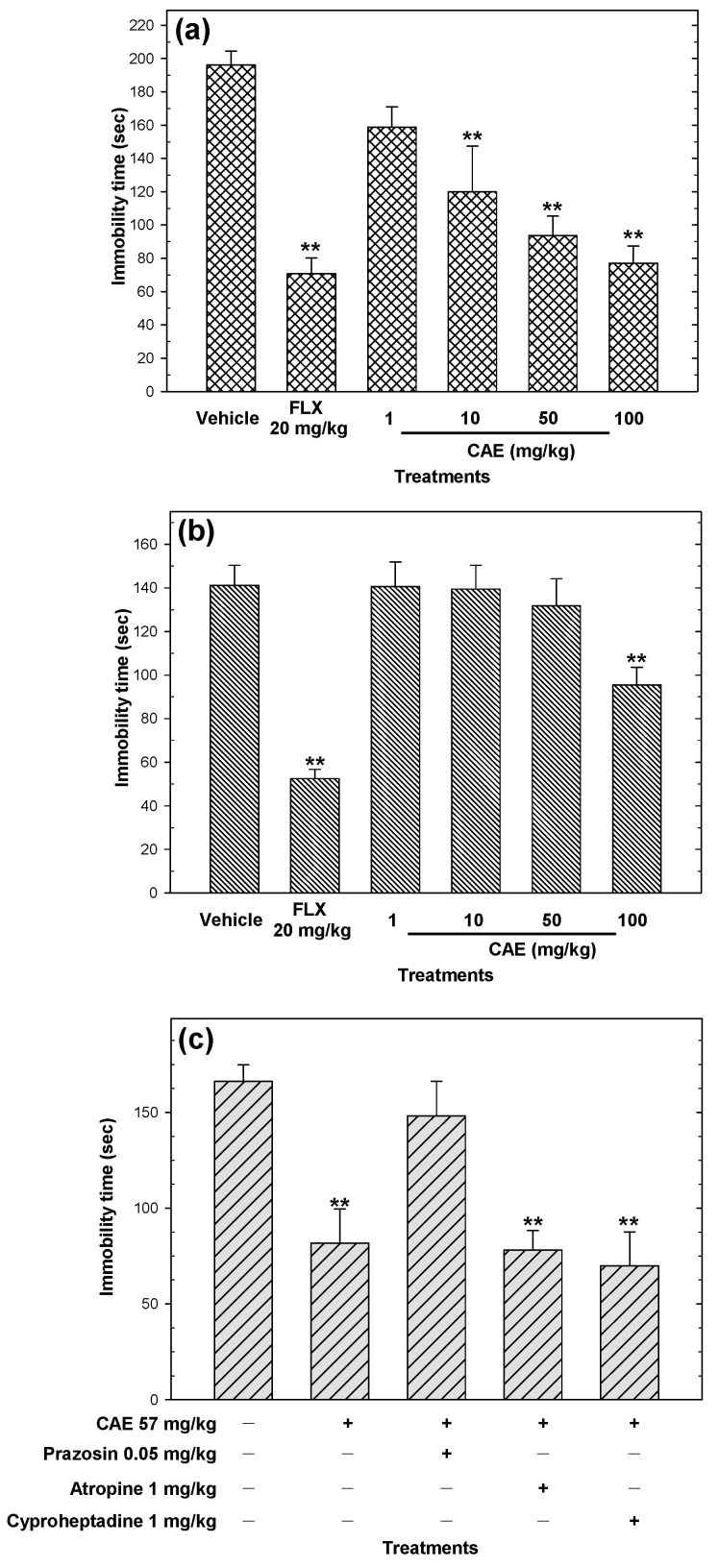
Antidepressant-like actions of CAE (10–100 mg/kg p.o.) evaluating the time of immobility in the tail suspension test (**a**) and forced swimming test (**b**). The possible mechanism of action of CAE was evaluated in the tail suspension test (**c**). Fluoxetine (FLX) at 20 mg/kg p.o. was used as the positive control. Bars represent mean values (± SEM) for the experimental group. n = 8, ** *p* < 0.05 compared to the vehicle group.

**Figure 7 pharmaceuticals-15-01580-f007:**
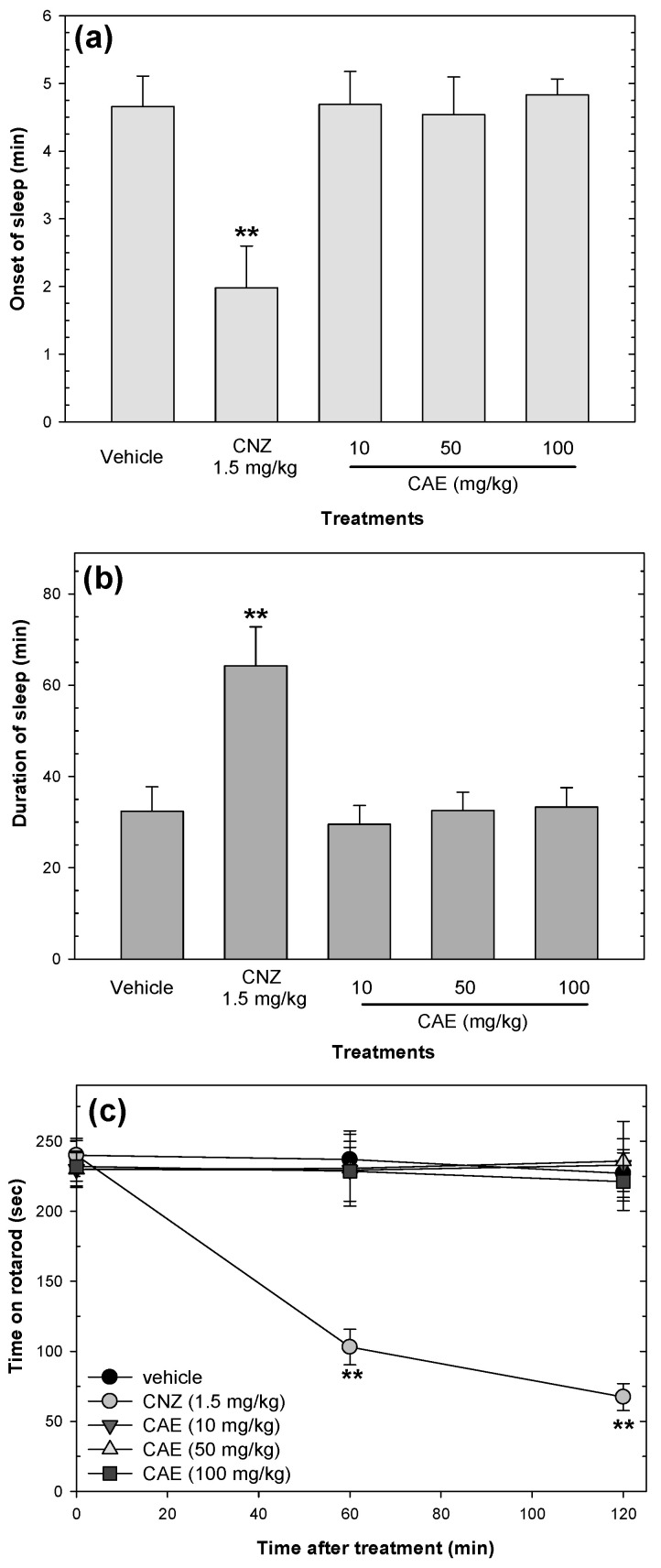
Sedative and motor coordination effects of CAE (1–50 mg/kg p.o.). The sedative effects of CAE were measured by the onset of sleep (**a**) and the duration of sleep (**b**) in the pentobarbital-induced sleep test, and the effects of CAE on motor coordination were evaluated with the rotarod test (**c**). Clonazepam (CNZ) at 1.5 mg/kg p.o. was used as the positive control. Bars (**a**,**b**) and shapes (**c**) represent mean values (± SEM) for the experimental group. n = 8, ** *p* < 0.05 compared to the vehicle group.

**Figure 8 pharmaceuticals-15-01580-f008:**
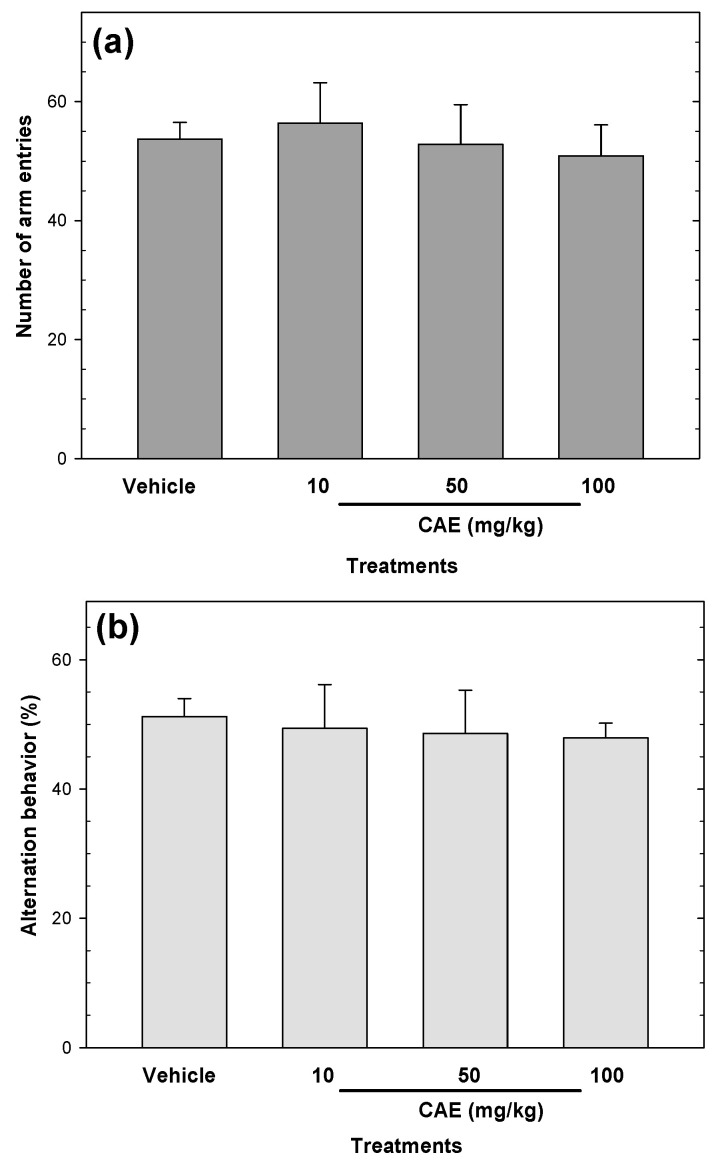
The effects of CAE on spatial working memory were evaluated by spontaneous alternation behavior (**a**) and the number of arm entries (**b**) measured during an 8 min session in the Y-maze test. Bars represent mean values (± SEM) for the experimental group. n = 8, ** *p* < 0.05 compared to the vehicle group.

**Figure 9 pharmaceuticals-15-01580-f009:**
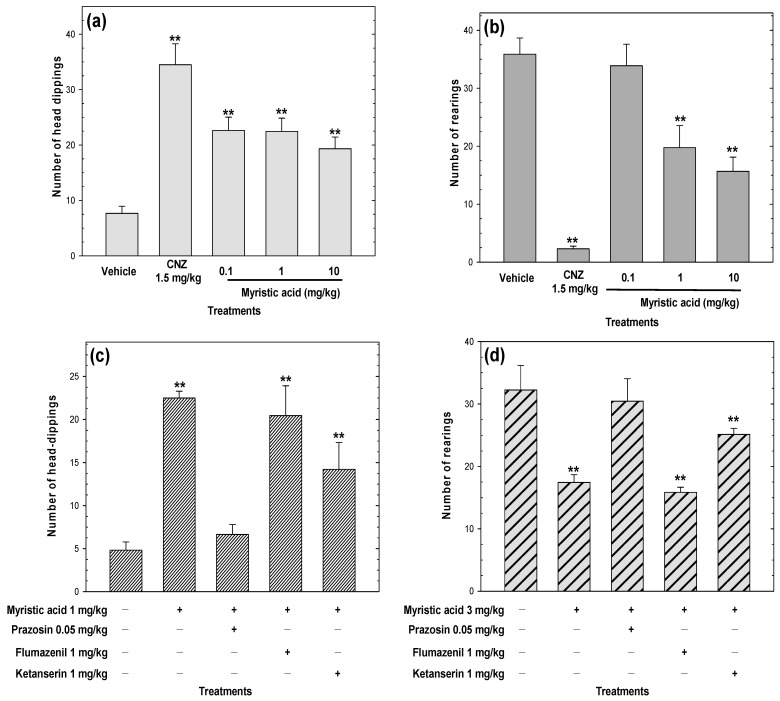
Anxiolytic-like effects of myristic acid (0.1–10 mg/kg p.o.) were measured in the cylinder exploratory test (**a**) and the hole-board test (**b**) during a 5 min exposure. The possible mechanism of action of myristic acid was evaluated in the cylinder exploratory test (**c**) and the hole-board test (**d**). Clonazepam (CNZ) at 1.5 mg/kg p.o. was used as the positive control. Bars represent mean values (± SEM) for the experimental group. n = 8, ** *p* < 0.05 compared to the vehicle group.

**Figure 10 pharmaceuticals-15-01580-f010:**
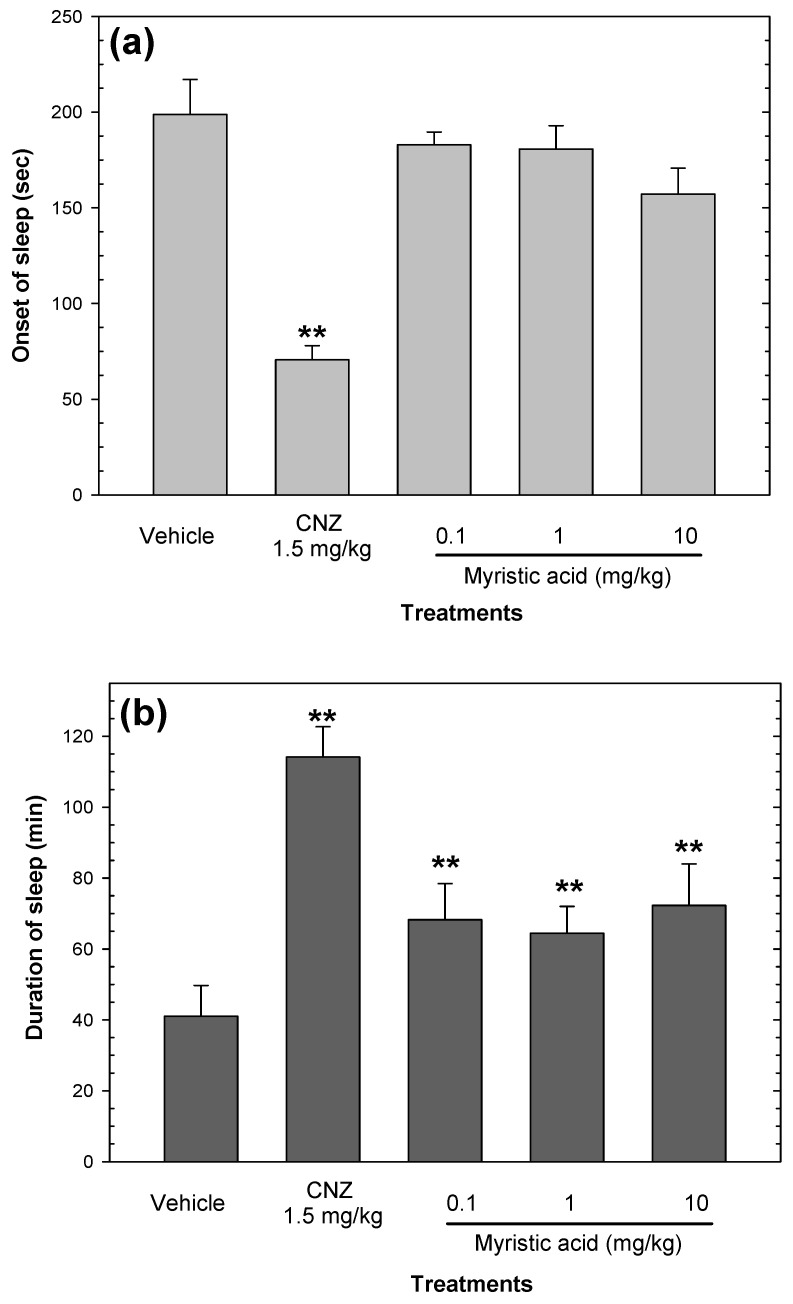
Sedative effects of myristic acid (0.1–10 mg/kg p.o.). The sedative effects of myristic acid were measured by the onset of sleep (**a**) and the duration of sleep (**b**) in the pentobarbital-induced sleep test. Clonazepam (CNZ) at 1.5 mg/kg p.o. was used as the positive control. Bars represent mean values (± SEM) for the experimental group. n = 8, ** *p* < 0.05 compared to the vehicle group.

**Figure 11 pharmaceuticals-15-01580-f011:**
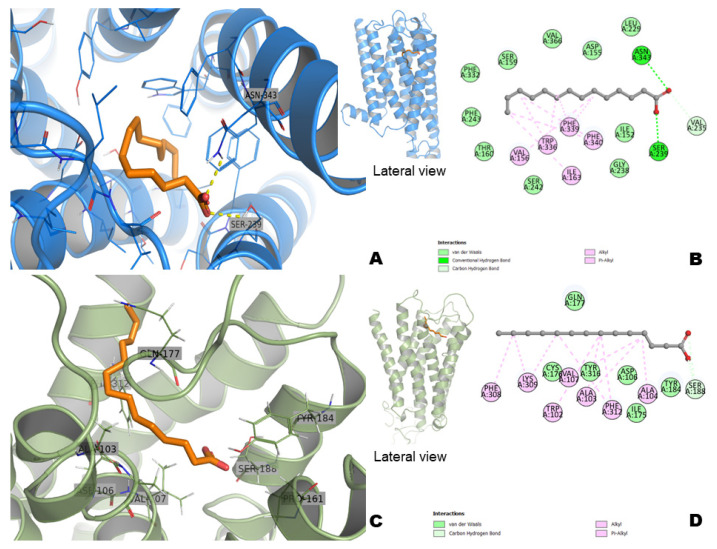
The docking complexes of myristic acid with 5-HT2A (**A**,**B**) and α_1A_-adrenergic receptor (**C**,**D**). Myristic acid is shown in stick representation (orange color). The elements are colored as follows: oxygen, red; nitrogen, blue; carbon, marine color to 5-HT2A and smudged color to α_1A_-adrenergic receptor. Dashed lines (yellow color) represent the hydrogen bonds and van der Waals interactions (pink and pale green colors).

**Table 1 pharmaceuticals-15-01580-t001:** Main compounds found in CAE.

No	Compound	RT	ERI	TRI	%RSD
1	Trimethylsilyl 3,4-bis[(trimethylsilyl)-oxy]butanoate	7.97	1422	1457	2.40
2	2,2,9,9-Tetramethyl-5,6-bis[(trimethylsilyl)oxy]-3,8-dioxa-2,9-disiladecane	8.673	1488	1478	0.68
3	meso-Erythritol, tetrakis(trimethylsilyl) ether	8.758	1496	1500	0.27
4	Tetronic acid, tetrakis-O-(trimethylsilyl)-	9.126	1531	1518	0.86
5	Pentane-1,2,3,5-tetraol tetraTMS dec	9.223	1540	1526	0.92
6	1,2,3,4-tetrakis-o-(trimethylsilyl)pentopyranose	9.847	1600	1602	0.12
7	6-Deoxy-2,3,4,5-tetrakis-O-(trimethylsilyl)hexose	9.969	1612	1632	1.23
8	Trimethylsilyl p-(trimethylsilyloxy)benzoate	10.083	1624	1621	0.19
9	arabinonic acid, 2,3,5-tris-o-(trimethylsilyl)-, γ-lactone	10.098	1626	1647	1.28
10	D-Arabinopyranose, tetrakis(trimethylsilyl) ether (isomer 2)	10.154	1632	1636	0.24
11	L-(-)-Arabitol, pentakis(trimethylsilyl) ether	10.842	1704	1746	2.41
12	1,2,3,4,5-Pentakis-O-(trimethylsilyl)pentitol	11.27	1751	1746	0.29
13	(2-(3,4-Bis[(trimethylsilyl)-oxy]phenyl)ethoxy)(trimethyl)silane	11.327	1757	1773	0.90
14	Vanillic acid, trimethylsiloxytrimethylsilyl ester	11.352	1760	1750	0.57
15	L-(-)-Arabitol, pentakis(trimethylsilyl) ether	11.51	1751	1746	0.29
16	D-(-)-Fructofuranose, pentakis(trimethylsilyl) ether (isomer 1)	11.613	1788	1792	0.22
17	D-Psicofuranose, pentakis(trimethylsilyl) ether (isomer 1)	11.617	1788	1799	0.61
18	D-(-)-Fructofuranose, pentakis(trimethylsilyl) ether (isomer 2)	11.68	1796	1800	0.22
19	D-(-)-Fructopyranose, pentakis(trimethylsilyl) ether (isomer 1)	11.741	1802	1802	0.00
20	Ethyl 4-(2-Hydroxymethylphenyl)butanoate	11.811	1810	1814	0.22
21	Benzoic acid, 3,4-bis[(trimethylsilyl)oxy]-, trimethylsilyl ester	11.831	1812	1826	0.77
22	Tetradecanoic acid, trimethylsilyl ester	12.12	1846	1842	0.22
23	D-Psicose, pentakis(trimethylsilyl) ether	12.351	1873	1884	0.58
24	D-(+)-Galactopyranose, pentakis(trimethylsilyl) ether (isomer 2)	12.401	1879	1889	0.53
25	3-(2-Ethyl-6-methylphenyl)-2-hydroxy-2,4-dimethyloxazolidine	12.491	1890	1874	0.85
26	Trimethylsilyl 3,5-dimethoxy-4-(trimethylsilyloxy)benzoate	12.514	1892	1884	0.42
27	D-Sorbitol, hexakis(trimethylsilyl) ether	12.701	1914	1919	0.26
28	Methyl 2,3,4,6-tetrakis-O-(trimethylsilyl)hexopyranoside	12.751	1920	1928	0.41
29	Hexadecanoic acid, trimethylsilyl ester	13.763	2026	2039	0.64
30	Myo-Inositol, 1,2,3,4,5,6-hexakis-O-(trimethylsilyl)-	14.027	2047	2053	0.29
31	Trimethylsilyl heptadecanoate	14.527	2086	2087	0.05
32	Linoleic acid trimethylsilyl esterI	15.038	2218	2201	0.77
33	cis-13-Octadecenoic acid, trimethylsilyl ester	14.088	2238	2232	0.27
34	Stearoxytrimethylsilane	15.268	2297	2277	0.88
35	2H-Pyrazino[1’,2’:1,5]pyrrolo[2,3-b]indole-1,4(3H,5aH)-dione, 11a-ethoxy-6,10b,11,11a-tetrahydro-10b-hydroxy-2,6-dimethyl-3-methylene (5aα,10bα,11aβ)	16.305	2654	2726	2.64
36	(E/Z)-5-(N.N-Diethylcarbamoylmethylene)-1-phenyl-3-phenylamino-2,5-dihydropyrrol-2-one	17.493	3062	3120	1.86
37	Sucrose, octakis(trimethylsilyl) ether	17.813	3172	3252	2.46
38	6H-Isoquino[2,1-b][2,7]naphthyridine, methanone derivative	17.95	3219	3266	1.44
39	D-(+)-Trehalose, octakis(trimethylsilyl) ether	18.492	3560	3556	0.11
40	Docosanoic acid, 13-[[2,3,4,6-tetrakis-O-(trimethylsilyl)-D-glucopyranosyl]oxy]	20.432	4072	4108	0.88

RT—retention time in minutes, ERI—experimental retention index, TRI—theoretical retention index, %RSD—relative standard deviation calculated for ERI.

## Data Availability

Data is contained within the article.

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
