# Peer review of "Neuropharmacological Activities of Ceiba aesculifolia (Kunth) Britten & Baker f (Malvaceae)"

_pharmaceuticals, 2022, doi:10.3390/ph15121580_

Round 1

Reviewer 1 Report

Dear authors

The Manuscript entitled "Neuropharmacological activities of Ceiba aesculifolia (Kunth) Britten & Baker f (Malvaceae)" It seems to be interesting because it does not have any pharmacological activity of the CAE extract. The doses used could have been based on experiments on plants of the same genus and from that research you could have used the doses to base on your experiments. Depending on what they found, the authors could have increased or decreased the doses. And then, the authors wouldn't even need to have done the LD50 of the extract.

The manuscript has experiments that evaluate behavioral tests, standardizing doses for 1, 10, 50 and 100 mg/kg, but in some experiments they do not use the dose of 100 mg/kg. The authors evaluate the mechanism of action, which makes the manuscript interesting, but it has a poor discussion of the findings of this mechanism. Peharps the authors could to write in title like "Evaluation of the mecanism of action of Ceiba aesculifolia (Kunth) Britten & Baker f (Malvaceae) through neurophamacological experiments" It will become more attractive to readers. The authors describe the methodology well summarized and some parts do not have a citation of where the experiments were based. I think material and methods is where you could go into more detail in the manuscript. Acute test in the results the authors say that it took 14 days and it is written in the methodology that it took 7 days. There are graphs in the results that need to be redone. The routes of administration of the positive standard drugs are not shown in the legends

In the elevated plus maze test, the authors do not show the data in the closed-arm graphs.

I didn't understand the difference in figure 5 between graph a and b

Discussion could be improved. The authors could discuss the results in more depth, as it stands it seems to be just a repetition of what was found in the results. Perhaps with further data discussed, the authors increase the number of references. Some of the material experiments and methods are not referenced, this would help increase references and make the manuscript more robust

Reviewer 2 Report

The authors described Neuropharmacological activities of Ceiba aesculifolia extract. I think the authors clearly demonstrated CAE pharmacology with rodent model, but the CAE has already been used and known as mood disorder drug. I feel this manuscript missed rationale evidence of either target protein or target by compound. I recommend that the author should carefully delineate what compound may affect the main target. I think just comparing on exiting drug effect is not enough for many readers.

Reviewer 3 Report

حفظ الترجمة

Dear author

It is important to value a scientific work in order to publish in a journal, so this article needs some modification of the corrections before publishing, below you will find the comments which must add:

Abstract

This section must be further developed, it is necessary to present the results in all the tests carried out in this study.

Introduction

It is necessary to develop this section with a general presentation on phytotherapy or bioactive compounds of natural origin. And also, the presentation of the species must be followed by this presentation.

Lin 40-43: a reference must be added

Results

Table 1: add the abbreviation (TR, ERI…) and remove “case” in the peak 40, and add the identification total (%).

2.2 Acute toxicity

Acute toxicity makes it possible to evaluate the toxicity of a natural compound, but it is necessary to add the behavior of the animals (immobility, diharrée, aggressive, food…). So the toxicity is not to determine LD50 but it is necessary to observe the behavior of the animals.

Discussion

It is necessary to add a small comparison with other plant of the same species, especially the GC/MS analysis.

Reference

references must be written in Journal format

Round 2

Reviewer 1 Report

The manuscript improved quality with additional experiments and with the corrections in the text. Well done!